# A Practical Guide to Therapeutic Drug Monitoring of Biologic Medications for Inflammatory Bowel Disease

**DOI:** 10.3390/jcm10214990

**Published:** 2021-10-27

**Authors:** Byron P. Vaughn

**Affiliations:** Inflammatory Bowel Disease Program, Division of Gastroenterology, Hepatology and Nutrition, University of Minnesota, Minneapolis, MN 55455, USA; bvaughn@umn.edu

**Keywords:** infliximab, adalimumab, vedolizumab, ustekinumab, Crohn’s disease, ulcerative colitis

## Abstract

Therapeutic drug monitoring (TDM) is a useful strategy to optimize biologic medications for inflammatory bowel disease not responsive to standard dosing regimens. TDM is cost effective for anti-tumor necrosis factor agents in the setting of loss of response (reactive TDM). Optimizing drug dosing when patients are in remission (proactive TDM) may be beneficial in certain circumstances. However, frequently the serum drug concentration in isolation becomes the focus TDM. Additionally, the lines of reactive and proactive TDM can quickly blur in many common clinical settings. Physicians employing a TDM based strategy need to place the drug concentration in context with the inflammatory status of the patient, the underlying pharmacokinetics and pharmacodynamics of the drug, the risk of immunogenicity, and the therapeutic goals for the patient. Physicians should understand the limits of TDM and feel comfortable making therapeutic decisions with imperfect information. The goal of this narrative review is to provide a framework of questions that physicians can use to employ TDM effectively in practice.

## 1. Introduction

“What levels do you aim for?” This might be the most consistently asked question at any modern inflammatory bowel disease (IBD) conference. It is the wrong question. The goal of using biologic therapy in IBD is to suppress inflammation (i.e., pharmacodynamics or what the drug does to the body). Biologic therapies approved by the US FDA at the time of this writing include monoclonal antibodies to anti TNF-α (infliximab, adalimumab, certolizumab, and golimumab) along with their related biosimilars, agents targeting leukocyte trafficking (the anti-integrin α4β7 monoclonal antibody vedolizumab) and monoclonal antibodies binding the p40 subunit of the pro-inflammatory interleukins (IL)-12 and-23 (ustekinumab). All current biologic therapies have assays in clinical use to measure the serum drug concentration and anti-drug antibody (ADA) concentration. Overreliance on targeting a serum concentration (i.e., pharmacokinetics) without an accurate clinical and inflammatory context will leave providers confused and patients underserved. “Targeting a level” for clinical decision-making is similar to a platitude, a flat truth. On one level, it is an easy to measure endpoint; but focusing only on the concentration belies the difficult, complex decisions involving IBD therapy. Serum drug concentrations are one of many important data points. Using serum drug concentrations appropriately requires knowledge of drug mechanism, the risk of immunogenicity, the inflammatory status of the disease, and realistic therapy goals.

The fact that therapeutic drug monitoring (TDM) is complex, should not dissuade the provider from the practice. Rather, with a few well-posed questions, this strategy can be easily implemented in practice. There are a number of excellent reviews that summarize the literature on TDM in IBD [1,2]. The goal of this review is to provide a useful framework for implementing the literature in practice. Providers should ultimately feel more confident in a TDM based strategy and less confident in any specific serum concentration. If “what levels do you aim for?” is the wrong question, then what is the right question? This review sets out to outline the right question(s) to effectivity use TDM in practice.

## 2. Association versus Causation

Drug concentrations are inversely associated with inflammation. In almost all post-induction studies, serum drug concentrations are significantly higher in patients who are in remission [3,4,5]. Ongoing inflammation is associated with lower drug concentrations. One important reason for this association is the protein losing colopathy seen with severe intestinal inflammation [6,7]. As the molecules constituting biologic therapies are roughly the same size (100–150 kilodaltons), this phenomenon is likely true for biologics as a class of therapy. There are other mechanisms for increased biologic metabolism. In states of inflammation monoclonal antibodies undergo increased proteolytic degradation [8]. These effects occur with a number of proteins that are negatively associated with inflammation, albumin being the most common. Albumin likely acts as a surrogate marker for monoclonal antibody turnover.

Therefore, on a population level, any cross-sectional analysis will have a trend of higher drug concentrations with increased remission rates. Importantly, these do not provide insight into a threshold effect (i.e., a limit at which higher concentrations would not decrease inflammation further). It is appealing to identify of a threshold effect, but the study designs prohibit that type of interpretation. It is problematic to use population level associations as actionable data on an individual level. For a given individual, a biologic concentration threshold may exist; but population pharmacokinetic studies do not provide that information. For example, a given cohort of IBD patients, infliximab concentrations >4 μg/mL may be associated with remission. However, specific individuals in that cohort will vary. A given individual with severe penetrating Crohn’s disease may only achieve remission once the infliximab concentration is >15 μg/mL. Or perhaps, infliximab is targeting the wrong inflammatory pathway in this that patient. Population association studies are important and hypothesis generating, but they should not be over-relied on to identify thresholds for individual patients in clinical practice.

## 3. What Is the Target?

The ideal target would directly inform the activity of the inflammatory pathway involved in the biologic mechanism of action. Rather than make changes based on a drug concentration, changes would be made based on a direct measure of the targeted inflammatory pathway. Some individuals would have pathway suppression at a low concentration, while some would require higher doses. However, the drug concentration itself is secondary to the measurement of the pathway. In this theoretical model, measurements of inflammatory pathways would also help guide biologic therapy choice.

Unfortunately, there is no current measurement in clinical practice of a specific inflammatory pathway. When considering TDM it is essential to first establish inflammation and use inflammation as the target over time [9,10]. While our measure of inflammation are not pathway specific, trends over time can aide in assessing a therapeutic response [11]. Low drug concentrations typically also predict who will respond to a dose intensification [12,13]. The combination of active inflammation and low drug concentration suggests that the inflammatory pathway is not saturated, and dose escalation is likely to capture or recapture a response. However, some biologic concentrations may be less predictive of response than others. Typically the cytokine based biologic therapies (anti-TNFs and anti-IL-12/23) have a high correlation between concentration and response to dose escalation [12,13], while the anti-integrins may not [14]. Other factors to consider in how helpful a drug concentration will be in predicting a response to dose intensification include the current dose/frequency, the underlying risk of immunogenicity, and if the patient had a primary response. An overview of key questions to consider with TDM is found in Figure 1.

## 4. Reactive Therapeutic Drug Monitoring for Secondary Loss of Response in the Maintenance Phase of Biologic Therapy

Patients with an initial clinical response who then lose response in the maintenance phase with recurrence of inflammation are likely benefit from a therapeutic dose escalation in two settings: (1) lack of anti-drug antibodies (2) low serum trough concentrations.

### 4.1. Anti-Drug Antibodies

Anti-drug antibodies (ADAs) effectively eliminate a given biologic and can also lead to infusion or injection related reactions [15,16]. The reference range of ADA detection varies with different assays. Most assays in clinical use are drug intolerant, meaning ADAs are only detectable when low or absent serum drug concentration. Compared to drug intolerant assays, drug tolerant assays (i.e., assays that can detect ADAs in the presence of serum drug) appear to be more consistent in ADA detection [17]. Some ADAs are clinically significant, while other appear to be transitory [18]. Non-neutralizing ADAs may bind the drug without inhibiting the pharmacologic effect [19]. An ADA is likely neutralizing in the setting of high ADA concentrations with low/absent serum drug. If the measurement is a trough measurement and the drug concentration is low/absent with ADA, then the ADA is likely neutralizing the drug. A drug concentration of zero is clearly the easiest to interpret, although understanding the limitations of the specific assay being used remains important. My research group previously identified a subset of individuals with positive ADA to adalimumab and a drug concentration of zero who had a detectable drug concentration after dose escalation, even though ADAs persisted [20]. In these settings it is imperative to closely follow the clinical course for changes in inflammation and drug related side effects.

Low level ADAs, particularly if using a drug tolerant assay, are common and often disappear over time [21]. These likely represent clinically insignificant, non-neutralizing antibodies. Addition of an immunomodulator can eliminate these ADAs [22,23]. High antibody concentration with a non-low drug concentration may reflect lab error. In these cases, repeating the drug concentration and ADA with a different assay (e.g., drug tolerant) may provide more data. Ultimately, clinical context is still required. Table 1 summarizes the likelihood of neutralizing antibodies based on the serum drug and anti-drug antibody concentration.

In the setting of recurrent inflammation with low/absent serum trough concentrations, and high ADAs, the best choice is to stop the current therapy and proceed to another biologic. The provider must consider the future risk of anti-drug antibodies with another biologic therapy. Patients who develop ADAs to one biologic are more likely to develop repeat ADAs to another biologic [24]. Consideration should be given to adjunct therapy with an immunomodulator to prevent ADAs, [25] using a biologic with a baseline low immunogenicity rate, or using a non-biologic therapy to avoid the concern for ADAs all together.

### 4.2. Low Trough Concentrations

Active inflammation may improve with a biologic dose escalation in the presence of low serum trough concentrations. However, there is no universal definition of low. There is substantial variation in an individual’s drug concentration-clinical response, making id challenging to identify universal concentration threshold values. The best data regarding a target concentration are from the TAXIT study, a randomized controlled trial of a TDM strategy versus empiric dose escalation for patients with Crohn’s disease on infliximab. In the optimization phase, patients with Crohn’s disease were more likely to be in remission after a dose escalation for infliximab trough concentrations under 3 μg/mL [26]. However, using other outcomes different infliximab cut-offs can be identified: 3–5 μg/mL for clinical disease, [27] 8–12 μg/mL for endoscopic remission, [28] and 18–20 μg/mL for fistula healing [29,30]. As noted above, low trough concentrations from population cohort studies are problematic as they do not inform an individual’s inflammatory pathway activity. The concentration is not the target. The true target is suppression of an inflammatory pathway, which we cannot effectively measure. The appropriate clinical question is not “*what drug level should I aim for*”, but “*will this patient’s inflammation regress with a dose escalation?*” There is no simple answer.

The main benefit to reactive-TDM is in the identification of ADAs. Sparing a patient any dose of a biologic that is no longer effective is beneficial and cost effective [31,32,33]. However, in the absence of ADAs it is unclear what the upper limit of a serum drug concentration should be. If on standard maintenance dosing without ADAs, dose escalation should strongly be considered. For patients already dose escalated, but with still low levels, there are little data to guide. Patients with concentrations under 10–12 μg/mL (for anti-TNFs) may still benefit from dose escalation. Some early data suggest that trough concentrations have little effect on predicting dose escalation to vedolizumab [14,34]. Ultimately, having objective measures of inflammation, and prespecified time points to assess a change in inflammation (6–12 weeks typically) are key to determining success or failure of a dose change.

### 4.3. Subclinical Inflammation

Until now, no distinction has been made between inflammation causing clinical symptoms and ongoing intestinal inflammation without any gastrointestinal symptoms, i.e., subclinical inflammation. In the setting of subclinical inflammation, therapy optimization based on trough concentrations can be quite helpful. This is essentially “reactive TDM”, although with some important nuance. Consider the following scenario. Biologic A gets a patient into clinical remission, but the patient has ongoing endoscopic inflammation. The provider checks a trough concentration of biologic A, which returns in the “therapeutic” range. In typical reactive TDM (i.e., with clinical symptoms from inflammation), changing biologics to a different mechanism would be appropriate. However, in this case, presumably the clinical remission is due to biologic A. Changing to therapy B could lead to a clinical disease recurrence. This is not to say that changing biologic therapy is wrong. Changing to therapy B could lead to both clinical and endoscopic remission. Or increasing the dose of biologic A further could result in complete suppression of inflammation. In these situations, a discussion with the patient on the risks and benefits is essential. The decision should include how many other therapies have been tried, the risks of ongoing inflammation (e.g., colon cancer in the setting of PSC), and the patient’s treatment goals. A holistic approach to an individual can help determine the need for treatment modification. For example, certain comorbidities (e.g., venous thromboembolism) or extra-intestinal manifestations of IBD (e.g., arthritis) may sway a provider to escalate or change a biologic despite the lack of GI symptoms. Managing subclinical inflammation does not lend itself well to an algorithm, and falls more into an art.

## 5. Reactive Therapeutic Drug Monitoring during Induction Therapy

The concept of “primary non-response” is simple on face value: A medication fails to control inflammation because it is not targeting the correct inflammatory pathway(s). Unfortunately, as noted previously, we do not have a measurement what inflammatory pathways are overactive in an individual patient; nor do we have a measure of if that pathway responded to a given therapy. Thus, two problems exist with primary non-response: (1) was the therapy underdosed and (2) was enough time given to see a benefit. TDM may help a provider determine if a primary non-responder is due to underdosing of a biologic therapy.

To date, drug trials in IBD use fixed dosing strategies for large, randomized placebo-controlled studies. For medications with reproducible pharmacokinetics, this process works well. However, with biologic therapy, there is wide individual variation in the serum concentration that is also dependent on the disease. In cases of acute-severe colitis, infliximab loss in the stool through a protein losing colopathy is well documented [6]. The disease process has a negative feedback loop on the drug itself. As such, there are observational data that accelerated induction dosing of infliximab for acute, severe ulcerative colitis decreases the early colectomy rates [35]. However, increasing all induction dosing is not likely to be an effective strategy. A recent study of high dose adalimumab versus standard dose adalimumab did not improve post induction remission rates for Crohn’s disease or ulcerative colitis [36,37]. These are important studies, yet they are not evaluating TDM in induction. The goal of TDM in induction is to identify the subpopulation that is most likely to benefit from escalated induction dosing.

An induction TDM strategy presents some practical problems. First, and most importantly, there is a dearth of data on drug concentrations in the induction phase. An extensive review by Sparrow et al. thoroughly summarizes the post induction concentration data to date [38]. The authors note the current data are from observational studies and post-hoc analysis of clinical trials. While this is currently a knowledge gap, there exist several ongoing trials incorporating TDM into the induction period (e.g., NCT03029143). Second, the turnaround time for the results of drug concentration testing can make dose adjustments in the induction phase difficult. Most clinicians do not have “in-house” drug concentration assays and rely on a runaround time of a week or more for the information to return. Fortunately, ADAs are less common in the induction phase. Therefore, empiric dose escalation may be a more useful clinical strategy. Risk calculators or dashboards exist and are being developed (e.g., NCT04835506) aid in induction TDM decisions [39,40].

A recent trial of randomized controlled trial of proactive TDM dosed infliximab versus standard dosed infliximab in Norway (NOR-DRUM Part A) did not identify a benefit of clinical remission across a number of inflammatory conditions (including IBD) [41]. This trial highlights the importance of clinical context paired with TDM. The trial protocol had strict escalation and de-escalation criteria to achieve infliximab concentrations in a pre-set therapeutic range, regardless of inflammatory status. While “proactive”, this approach is not “personalized”. Strict therapeutic ranges without the context of the inflammatory status are not likely to be useful strategies.

Overall, reactive TDM in the induction phase is like TDM in the maintenance phase. On standard dosing, if a patient is not responding to a therapy, the goal of TDM is to identify anti-drug antibodies. Without anti-drug antibodies, it is likely that standard dosed biologic therapy should be escalated.

## 6. Therapeutic Drug Monitoring When a Patient Is in Deep Remission

In the absence of inflammation or a suspected antibody mediated side effect, TDM does not have a direct clinical benefit to a patient. Meaning the knowledge of the serum drug concentration and ADA status and subsequent therapeutic changes will not immediately improve a patient’s quality of life. The most important role of TDM in this setting is the identification of anti-drug antibodies. Potentially 15% of patients with a stable clinical response or remission will have undetectable serum drug and positive ADAs to infliximab [26,27]. At this point the provider needs to determine if the patient should be de-escalated from biologic therapy completely or transition to a new therapy. The decision to perform TDM in deep should depend on the underlying risk of immunogenicity. If the risk of immunogenicity is quite small (no prior history of ADA, patient on combination therapy with immunomodulator, or using a biologic with low ADA rate), then the benefit of TDM is low. Likely the greatest benefit of TDM in deep remission is with anti-TNF monotherapy.

## 7. Special Situations

### 7.1. Reinduction of a Biologic Therapy

Re-starting a biologic runs the risk of inducing anti-drug antibodies [42]. The greatest risk of infusion reaction is with the second re-induction dose. Identification of anti-drug antibodies after the first dose can spare a patient an infusion reaction with the second dose [43]. A practical way to implement this strategy is to alter the induction dose for infusions. For infliximab, an induction schedule could be at week 0, 4, and 8. A serum drug concentration and antibody status at week 2 can inform the presence of antibodies and determine subsequent dosing. For self-injectable biologic therapies, it is not possible to alter the induction schedule in the same way. Obtaining a drug level 1–2 weeks after the first dose is helpful, but given the typical turnaround time, the information may not be back before administration of the second dose.

### 7.2. De-Escalation of Therapy

After a patient is in remission on combination therapy with a biologic therapy and immunomodulator, a provider may consider stopping the immunomodulator and continuing biologic monotherapy to maintain remission. This is a reasonable strategy to minimize long term side effects of dual immunosuppression. Immunomodulator cessation may reduce serum biologic concentrations. Obtaining a serum drug concentration while on combination therapy can help to guide de-escalation. The risk is the development of ADAs after stopping the immunomodulator. In the case of very low trough concentrations, providers may consider dose escalating the biologic prior to stopping the immunomodulator. Or at a minimum, repeating the serum drug concentration on monotherapy soon after de-escalation.

TDM may also aide in biologic de-escalation in the maintenance phase. Some patients can successfully de-escalate the dose of a biologic [44]. In the STORI trial, where patients on infliximab and a thiopurine stopped the infliximab, those with low trough concentrations (<2 mg/ul) were less likely to experience a clinical relapse [45]. Measuring both biologic and thiopurine concentration can help inform de-escalation. If one is “subtherapeutic” that may be the best to de-escalate [46]. Informing the patient on the potential long-term risks of therapy de-escalation is essential, and this strategy should be individualized with close follow-up.

### 7.3. Suspicion for Antibody Mediated Side Effects

ADAs can cause a number adverse clinical effects. Most common is disease recurrence in the setting of rapidly cleared drug. ADAs can also lead to infusion reactions, both acute and delayed [47]. Most data on TDM to guide dosing are based on trough concentrations. However, when the question pertains to a potential side effect, non-trough levels may be helpful. Knowledge of the drug and antibody assay is essential in these cases. As noted before, most assays in clinical practice can only detect ADAs when the serum drug concentration is very low or absent. In this case, drawing the drug concentration too early may fail to identify antibodies. For infusion-based biologics, waiting 4-weeks post infusion should be sufficient for clinically significant antibodies to clear the serum drug and be identified. For subcutaneous biologics, it is likely best to obtain a trough measurement. Alternatively, use of a TDM assay that can measure ADAs in the presence of circulating drug can circumvent this issue.

## 8. Conclusions

TDM is an extremely useful tool in the management of IBD patients on biologic therapies. However, TDM is only helpful in the context of the patient’s inflammatory status and response to therapy. Population pharmacodynamic studies can identify helpful trends in serum drug concentrations but should not be mistaken for individual therapeutic thresholds. Practically, clinicians should always objectively measure inflammation at the time of TDM. Clinicians should also have a working understanding of the risk of immunogenicity of a given biologic therapy in a patient. Finally, clinicians using a TDM based strategy should understand what question they are trying to answer before a drug concentration is measured (Figure 1). TDM should be used as a form of personalized medicine incorporating the clinical inflammatory status, therapeutic options, and individual goals of care. Rather than focus too much on *reactive* or *proactive* TDM, we should focus on *personalized* TDM.

## Figures and Tables

**Figure 1 jcm-10-04990-f001:**
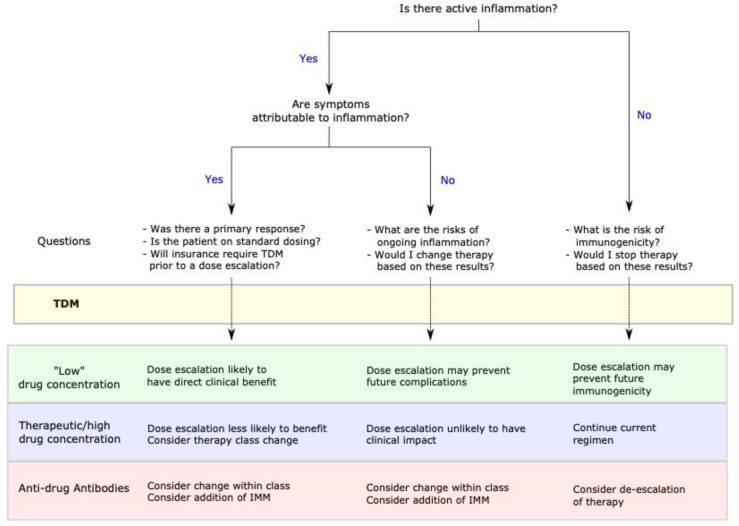
Algorithm of key questions to consider prior to implementing therapeutic drug monitoring (TDM) in clinical practice. Clinical benefit refers to immediate symptomatic improvement. Clinicians should have a clear understanding of the disease status prior to TDM and be able to articulate to the patient the benefit of a dose change in response to the natural history of the disease. In some cases of low-level antibodies, it may be appropriate to add an immunomodulator (IMM) to the current therapy to eliminate anti-drug antibodies. This choice should be individualized based on other available therapies and risks of dual immunosuppression for patient.

**Table 1 jcm-10-04990-t001:** Framework for evaluating detectable biologic trough concentration and detectable anti-drug antibodies.

		Serum Drug Concentration	
		“High” concentration	“Low” concentration
**Serum Anti-drug** **Antibody** **Concentration**	“High” antibodies	Neutralizing antibodies unlikely Consider lab error Consider repeating	Neutralizing antibodies likely Consider within class change
	“Low” antibodies	Neutralizing antibodies unlikely Consider repeating	Neutralizing antibodies possible Consider dose escalation or addition of IMM

IMM: Immunomodulator. Assuming trough concentrations at standard maintenance dosing, anti-drug antibodies in the presence of adequate or high serum drug concentration are unlikely to be neutralizing. Low serum drug concentrations increase the chance of neutralizing antibodies. Any antibody in the setting of probable drug mediated side effects may still be clinically significant.

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
