# Peer review of "A Practical Guide to Therapeutic Drug Monitoring of Biologic Medications for Inflammatory Bowel Disease"

_jcm, 2021, doi:10.3390/jcm10214990_

Round 1
Reviewer 1 Report
A very interesting paper about the possible use of therapeutic drug monitoring in inflammatory bowel disease patients treated with biological drugs; the main defect that, in my opinion, this paper has is the fact that it faces this argument not focusing on single drugs, but only cover the problem from a general point of view, not well differentiating among therapies.
The paper seems more a Commentary than a pure review; if so, the type of article could be changed; otherwise, a small materials and methods section stating what keywords were used and what databases were searched to select the studies included would be a great addition.
line 149-151 and line 163 you report serum drug concentration associated with IBD remission, but you do not specify the drug measured, but only the class; given the variability of dosage among the drugs in the same therapeutic class, the name of the referred drug should be specified.
In order to make this paper more interesting to the clinicians, I would probably expand conclusions giving more precise advice to clinicians.
In the introduction, you did not mention what are the currently available biologic drugs for IBD, so a paragraph like :"Biologic therapies include monoclonal antibodies anti TNF-α and their related biosimilars (infliximab
,adalimumab, and golimumab ), agents targeting leukocyte trafficking (the anti-integrin α4β7 vedolizumab ) and monoclonal antibodies binding the p40 subunit of the pro-inflammatory interleukins (IL)-12 and -23 (ustekinumab)." and adding a reference such as: doi: 10.1080/03007995.2020.1786681.
Author Response
Please see attachment "Responses to reviewer 1"

Reviewer 2 Report
This is a very clear and well written manuscript, summerized in a practical and clinically oriented way. I fully agree that TDM is a tool rather than a target.
I do have some minor requests for the author:
- Figure 1 (flow chart) - patients with active iniflammation and low or undetectable drug levels and ADA can also sometimes benefit from addition of immunomodulator (Ben Horin et al https://pubmed.ncbi.nlm.nih.gov/23103905/, Colman et al https://academic.oup.com/ibdjournal/article-abstract/27/4/507/5840484)
- Section of subclinical inflammation, line 183-184 - please add comorbidities and EIM as factors that should be taken into account in the decision making.
- In the context of early drug monitoring during induction it is worth mentioning the specific senario of ASUC in which there is some data to support early accelerated dosing - Gibson et al. CGH 2015
Author Response
Please see attachment "Responses to reviewer 2"
